# Family Welfare Expenditure, Contraceptive Use, Sources and Method-Mix in India

**Sheuli Misra** [1,*] , **Srinivas Goli** [1,2] , **Md Juel Rana** [3] , **Abhishek Gautam** [4] , **Nitin Datta** [4] , **Priya Nanda** [5] and **Ravi Verma** [4]

1   Centre for the Study of Regional Development, School of Social Sciences, Jawaharlal Nehru University (JNU), New Delhi 110001, India; srinivas.goli@uwa.edu.au
2   Australia India Institute (AII), UWA Public Policy Institute, Perth, WA 6000, Australia
3   International Institute for Population Sciences, Mumbai 400001, India; jranajnu@gmail.com
4   International Center for Research on Women (ICRW), New Delhi 110001, India; agautam@icrw.org (A.G.); ndatta@icrw.org (N.D.); rverma@icrw.org (R.V.)
5   Bill & Melinda Gates Foundation, India Country Office, New Delhi 110001, India; Priya.Nanda@gatesfoundation.org
*   Correspondence: sheulimisra93@gmail.com

**Abstract:** Making universal access to sexual and reproductive health care a reality, and thus building momentum for comprehensive family planning by 2030, is key for achieving sustainable development goals. However, in the last decade, India has been retreating from progress achieved in access to family planning. Family planning progress for a large country such as India is critical for achieving sustainable developmental goals. Against this backdrop, the paper investigated the question of how far family welfare expenditure affects contraceptive use, sources of contraceptive methods, and method-mix using triangulation of micro and macro data analyses. Our findings suggest that, except for female sterilizations, modern methods of contraception do not show a positive relationship with family welfare expenditure. Notwithstanding a rise in overall family welfare expenditure, spending on core family planning programs stagnates. State-wise and socio-economic heterogeneity in source-mix and method-mix continued to influence contraceptive access in India. Method-mix continued to skew towards female sterilization. Public sector access is helpful only for promoting female sterilization. Thus, the source-mix for modern contraceptives presents a clear public-private divide. Over time, access to all contraceptive methods by public sources declined while the private sector has failed to fill the gap. In conclusion, this study identified a need for revitalizing family planning programs to promote spacing methods in relatively lower-performing states and socio-economic groups to increase overall contraceptive access and use in India through the rise in core family planning expenditure.

**Keywords:** contraceptive use; family welfare expenditure; sources of contraception; method-mix; India

## 1. Introduction

Family Planning (hereafter FP) is often characterized as the "second-best buy" because of its cross-cutting impacts on human lives, both in the short and long term [1–5]. The benefits of Universal Access to Family Planning and returns of investment have impacted not only the lives of vulnerable groups but also contributed to the social and economic development in low and middle-income countries [6–10]. Thus, building momentum for comprehensive FP is key to making universal access to sexual and reproductive health care a reality, thereby achieving sustainable development goals (SDGs) by 2030. Yet, a considerable number of developing countries, particularly South Asia and Sub-Saharan Africa, are still unable to improve the level of modern contraceptive prevalence rate (mCPR), especially access to spacing methods [11,12]. According to UNFPA estimates for 2017, the lack of access to lifesaving FP methods has caused 67 million unintended

pregnancies and one-third of 303,000 maternal deaths in these countries [13]. While other latest estimates suggest that lack of FP casts a shadow over the future of around 218 million women of reproductive age in developing countries who have an unmet need for modern contraception [5].

The FP services in the majority of developing nations are predominantly provided by subsidized public sectors. Large numbers of users, especially from the lower-income quintiles, are more likely to opt for these sectors because of affordable or free services [14]. Although private sources (for-profit and not-for-profit organizations), social marketing organizations (SMOs), and other informal sectors are also playing a significant role in fulfilling the existing gaps of FP services, profitable private sources are often criticized by many as a reason for higher economic burdens and inequalities in service utilization [15,16]. Moreover, the method-mix of contraceptive use is highly skewed in developing countries, with more than half of all users being supported by just 1 or 2 methods. Such a pattern limits an array of user options and constrains the total prevalence of use, leading to unplanned pregnancies and births or abortions. Thus, the overall performance of sexual and reproductive health services in developing countries largely depends on government investments in FP services [17].

Although a few previous studies in a global context have suggested that public spending on FP significantly contributes to positive effects on FP use and reproductive, maternal, child, and adolescent health care (RMNCH+A) and outcomes [18–21], many developing countries narrowly understand FP's role as limited to mere fertility control. However, fertility limiting is just one of the several goals (e.g., timing and spacing of births and protection against sexual and reproductive tract infections, etc.) that FP accomplishes. Surprisingly, without considering FP's wider outcomes, many of these countries started disinvesting in the core FP programs (i.e., supply and services of FP) in the aftermath of a well-established path of fertility decline. Moreover, a fall in international donor financing for core FP programs has raised significant concern related to its sustainability, poor outreach, and quality of FP services that may have many stern long-term implications for meeting the unmet need for contraception and accomplishment of RMNCH+A outcomes in developing countries [1,6,22]. In turn, this can also have a major impact on wider developmental outcomes by delaying the achievement of the SDGs [5,23].

*Indian Context*

Along with 192 countries on the post-development agenda of sustainable development in 2015, India committed to achieving the SDG-3 of Universal Health Coverage by 2030 [24]. However, so far, India's progress is not satisfactory regarding the timely achievement of the universal access to sexual and reproductive health care services listed in SDG-3.7 [23,25–27]. With its historic initiation in the1950s, the national policy for the FP program in India has its channel for the supply of services mainly through the public sector. However, family welfare expenditure (FWE) was integrated into reproductive and child health (RCH) services during the mid-1990s when the intent of the national FP program was shifted from a "population control centric" approach to the "reproductive and child health" approach. As a consequence, emphasis on core FP expenditure started waning, which later influenced the progress of the contraceptive prevalence rate (CPR) [28,29]. During the time of declining or stagnating public financing of FP (1992–1993 to 2004–2005), the progress in CPR more or less stagnated. Although India re-emphasized FP services after the London Summit in 2012, the CPR has slightly declined even after the rise in overall FWE [30–33]. Moreover, a recent study also suggests that the out-of-pocket expenditure on female sterilization is as high as 70 to 79 percent notwithstanding highly subsidized or free maternity care programs in place [34].

However, the mystery of decline in contraceptive use among currently married women from 56 percentin 2005–2006 to 54 percent in 2015–2016 is still unresolved [31]. Although literature related to access, demand, method-mix, and quality of services in FP are abundant [26,27,35–40], the role of FP expenditure in determining sources of contraceptive

methods and method-mix are yet to be addressed. Some of the previous studies suggest that the private (commercial) sector in India can complement the public sector for FP services. Still, the road map to engage these two sectors remains a challenge and less understood [29,41]. Therefore, this study draws the dynamic relationship among the financing for FP programs mainly on public sector expenditure, sources, and use of modern contraceptive methods. In particular, this study has three objectives. The first is to assess the effects of FWE on the use of modern contraceptive methods at the macro level. Second, we examined the question of whether the rise in FWE increases the share of access to the public sector as the source of modern contraception in the country. Furthermore, we investigated whether there is any association between the public-private ratio (PPR) in contraceptive access and the use of the type of contraceptive methods? The third objective is using micro-level evidence, we examined the socio-economic and demographic factors associated with accessing the public sector as the source and type of modern contraceptive methods. With these three objectives and using both micro and macro-level empirical analyses based on successive rounds of National Family Health Surveys (NFHS), the study makes a significant contribution to existing evidence on progress achieved in family planning through the family welfare program in India. Findings are critical from a family welfare policy perspective.

## 2. Materials and Methods

### 2.1. Data Sources

The data was collected from different sources for the macro and micro-level analyses. The unit of analyses are 30 state and union territories of India and individual women aged 15–49 years for macro and micro data, respectively. The data on the FWE and public health expenditure (PHE) was taken from the yearbooks of the Ministry of Health and Family Welfare, Government of India, and converted from the current price to constant price (2011–2012) [31,42,43]. The study used data from the Census of India (2011) on the female population in the reproductive age group to estimate the per capita FWE. For micro-data analyses, the information on contraceptive use and its sources were collected from the four successive rounds of the NFHS, 1992–2016. NFHS is a part of the worldwide Demographic and Health Surveys (DHS), which provides information on essential indicators of FP along with a large number of demographic and health indicators [32,44–46]. These surveys were based on a two-stage systematic random sampling design applied both for rural and urban areas to produce separate estimates at the state level. For the analyses, our study used a sample of 84,558, 84,862, 87,925, and 499,627 currently married women aged 15–49 years from NFHS-I (1992–93), NFHS-II (1998–99), NFHS-III (2005–06), and NHFS-IV (2015–16), respectively. Details on sampling design and data quality from NFHS were reported elsewhere [32,44–46].

### 2.2. Empirical Strategy

The paper has a three-fold empirical methodology: (1) Descriptive analyses; (2) Macro data analyses (viz. correlations and panel data regression analyses); (3) Microdata analyses (e.g., binary logistic regression). Below, we have explained all three empirical approaches and statistical models applied in detail.

#### 2.2.1. Descriptive Analyses

Before conducting the main empirical analyses of the study, in the first stage, we described key variables of the study using descriptive statistics. We have presented outcome variables viz.public spending on FP, modern contraception use (spacing and limiting methods), the share of the public sector as the source of the modern contraceptive method, and the ratio of the public-private source of contraception by states and key socio-economic characteristics of women.

### 2.2.2. Macro Data Analyses: Variables and Model Description

In the second stage, the study assesses the effects of FWE on the use of modern contraceptive methods using macro-level data analyses. For this purpose, correlation and panel data regression models have been used to carry out the statistical exercise for achieving the objective of the effects of FWE on the use of modern contraceptive methods. The bivariate association of FWE and public-private ratio (PPR) with the level of modern contraceptive use was assessed using Pearson's correlation analysis. Since we have time-series data across 30 cross-sectional units (states and union territories), we have applied the panel data regression model.

The outcome variables are the prevalence of any modern method of contraception and spacing methods of contraception. In the case of any modern contraceptive methods, information was compiled from two questions: (1) "Are you currently using any method to delay or avoid getting pregnant?", and if yes, (2) "Which method are you using?" The modern methods were categorized into spacing and limiting methods. Out of ten modern contraceptive methods, eight were spacing methods, namely, pills, intrauterine devices (IUDs), injectables, female and male condoms, the lactational amenorrhea method, emergency contraception pills, and other modern methods. The rest of the two were limiting methods (female and male sterilization).

In the macro-level models, the two main predictors are the per capita FWE and the share of FWE in PHE. However, the models controlled for several state-level socioeconomic indicators: under-five mortality (U5MR), antenatal care visits (ANCs), the use of any modern methods of contraception, the use of any spacing methods, child marriage rate, the sex ratio at birth, female literacy rate, per capita net state domestic product (NSDP), the proportion of the urban population, scheduled caste/tribe (SC/ST), and Muslim population.

As stated above, the multivariate panel data regression models were carried out to estimate the effects of FWE on the outcome variables controlling the other covariates using the panel data. The "Hausman specification test" was applied to specify the fixed or random effect model for panel data regression analyses for a given outcome and predictor variables. Following Cameron and Trivedi [47], the statistical proofs of the fixed or random effect models were presented as follows:

Fixed effects model:

The Equation for the fixed effects model becomes

$$Y_{it} = \beta_1 X_{it} + \alpha_i + u_{it}$$

where—$\alpha_i$ ($i = 1, \ldots, n$) is the unknown intercept for each entity (*n* entity-specific intercepts). —$Y_{it}$ is the dependent variable (viz. modern contraceptive use, public-private source of contraception) where $i$ = entity and $t$ = time. –$X_{it}$ represents one independent variable (viz. log of per capita FWE, the ratio of FWE to PHE, female literacy rate, urban population share, per capita NSDP, etc.), —$\beta_1$ is the coefficient for that predictor variable, and —$u_{it}$ is the error term.

Random effects model:

$$Y_{it} = \beta_1 X_{it} + \alpha_i + u_{it} + \varepsilon_{it}$$

where—$\alpha_i$($i = 1, \ldots, n$) is the unknown intercept for each entity (*n* entity-specific intercepts). —$Y_{it}$ is the dependent variable (viz. modern contraceptive use, public-private source of contraception, etc.) where $i$ = entity and $t$ = time. —$X_{it}$ represents one independent variable (viz. log of per capita FWE, the ratio of FWE to PHE, female literacy rate, urban population share, per capita NSDP, etc.), —$\beta_1$ is the coefficient for that predictor variable, —$u_{it}$ is Between-entity error, and –$\varepsilon_{it,}$ is Within-entity error

### 2.2.3. Microdata Analyses: Variables and Model Description

In the third stage, for microdata analyses using NFHS data, a binary logistic regression model has been applied to assess whether the respondents are accessing the public sector

as the source of modern contraceptive methods. The outcome variable for the model is the public sector as the source of modern contraceptive methods. In NFHS, a range of options is provided for capturing the source of modern contraceptive methods used by the respondents at the time of the survey. The survey asks "Where did you obtain (a current method) the last time?" to gather information on sources of modern methods used by the respondents. Sources of modern contraceptives were clubbed into three categories, namely, public, private, and other. Public sources for contraceptive supply were categorized by aggregating all state-provided health facility centers like government/municipal hospital, government dispensary, UNC/UHP/UFWC, Primary Health Centre (PHC)/Community Health Centre (CHC)/rural hospital, sub-center/Auxiliary Nurse Midwifery (ANM), government mobile clinic/camp, Anganwadi/Integrated Child Development Services (ICDS) center, other public medical centers, other community-based workers/Accredited Social Health Activist (ASHA), family planning clinic, mobile clinic, government paramedic, and Employees' State Insurance (E.S.I) hospital. Private sources of contraceptives were measured from sources such as private hospitals, private doctor/clinic, private mobile clinic, vaidya/hakim/homeopath, traditional healer, dai/Traditional Birth Attendant (TBA), other private medical facilities, shop, husband/friend/other family members, and Non-Governmental Organization (NGO)/trust hospitals/clinics (field workers were combined and considered as a private source). While the remaining sources were grouped as the other category. After examining the sample distribution, the other category was excluded from the analyses due to the small sample size (less than 1 percent).

A set of socio-economic and demographic background characteristics of women have been included in the multivariate logistic regression model as control variables. Control variables include the place of residence (rural and urban), religion (Hindu, Muslim, or other), caste (SC, ST, OBC, or other), current age (15–21 years, 22–34 years, or above 34 years), age at first marriage (less than 15 years, 15–19 years, 20–24 years, 25–29 years, or 30 years and above), the level of education (illiterate, primary, secondary, or higher), son preference (son preference and no son preference), exposure to mass media (no exposure, partial exposure, or full exposure), household wealth status (poorest, poorer, middle, richer, or richest) and working status of women (not working or currently working).Son preferences among women were identified by having a preference for boys over girls and were stratified into son preference and no son preference. Women who were exposed to any mass media (radio/television/newspaper) were grouped into two categories, partial and full exposure; in contrast, those who did not have any exposure were considered as no exposure. Furthermore, to understand regional variations and individual state effects on study variables, we added a state variable in our analysis. North-Eastern states and Union Territories (UTs) were clubbed into two separate categories, North-East and Goa and UTs, to avoid the problem of small sample size and standard errors in the regression model. Moreover, for the sake of comparability in the pooled dataset, we merged Telangana with Andhra Pradesh (as it was created in 2014 from Andhra Pradesh).

As stated above, to identify the factors determining the choice of public sources for obtaining any modern contraceptive methods and the use of any modern spacing methods, the binary logistic regression model was applied. These models were estimated separately for combined pooled data of all four rounds of NFHS (1992–2016) and the latest rounds of survey data (2015–2016). Results were presented as odds ratios (OR) with a 95% confidence interval (CI). Below the statistical proofs of the model have been explained using FP from public sources as an outcome variable. For example, we could define the use of FP from public sources as

$$y_i \begin{cases} 1 & \text{if the } i\text{th FP from public sources} \\ 0 & \text{otherwise} \end{cases}$$

Following Retherford and Choe [48], for the above binary dependent variables ($y_i$), the binary logistic regression model can be written as

$$\text{Log}\left(\frac{P_{\text{FP from public sources}}}{1 - P_{\text{FP from public sources}}}\right) = \text{Logit}\left(P_{\text{FP from public sources}}\right) = b_0 + b_1 x_{1 = \text{types of FP methods}} + b_2 x_2 + b_3 x_3 + b_k x_k + e_k$$

$P_{\text{FP from public sources}}$ is the probability of receiving modern methods of contraception from public sources, $b_0$ is the *y*-intercept, and the term $b_1 x_1$ is the regression coefficient.

## 3. Results

This section has three parts. The first part deals with the description of the key indicators viz. expenditures on family welfare, contraceptive method-mix, and the dominance of public sector as the last source of modern contraception. The second part presents the results from the macro-level analyses which assess the effects of FWE on the use of modern contraception. The third part shows the socioeconomic and demographic determinants of the use of public sources as the last source of modern contraceptives using the micro-level data.

### 3.1. FWE and its Share in Total PHE

Figure 1 shows trends in total FWE at current and constant prices, as well as per capita expenditure for India between 1991–1992 and 2014–2015. The trends suggested that the total as well as per capita expenditure had slightly declined in the late 1990s and a moderate rise was observed after the 2000s. Similarly, the core FWE, presented as a share of FWE in total PHE, showed that public financing on FP had reduced over the period (1991–1992 to 2014–2015). It showed an 11 percent decrease in public sector financing at the national level, while the share of FWE was the lowest during 2004–2005. The absolute level FWE expenditure for the recent period shows a significant increase because the RCH-related expenditure was combined with FP as a part of an integrated approach.Consistent with the national pattern, our assessment also indicated a decrease in FWE in most of the states. The states which observed an increase in FWE over time were Arunachal Pradesh, Jammu and Kashmir, Himachal Pradesh, Mizoram, Rajasthan, Tripura, Tamil Nadu, and Uttar Pradesh (Table A1).

### 3.2. Contraceptive Method-Mix

In India and across the states, the contraceptive method-mix shows that about two-thirds of currently married women using female sterilization (67.2 percent in 2015–2016), and this is still the dominant FP method used for the last 24 years. But the use of male sterilization had decreased significantly from 8.5 percent in 1992–1993 to less than 1 percent in 2015–2016. Further, the use of IUDs also dropped from 5 to 3 percent at the same time. In contrast, the use of oral pills and condoms increased substantially from 3 to 6 percent and 6 to 10 percent, respectively, during 1992–1993 to 2015–2016. The state-wise trends and patterns of the contraceptive method-mix detect a diverse pattern. Compared to the national average, the South Indian states, along with Madhya Pradesh, Maharashtra, and Jharkhand, had a substantially higher percentage of female sterilization users. Relatively lower use of female sterilization was observed in North-Eastern states, West Bengal, Uttar Pradesh.It declined considerably in Odisha. Overall, the states traditionally known for higher use of female sterilization have continued to be on the higher side. In comparison, the states traditionally with lower use of female sterilization observed a further decline in it during the same time. A clear state-wise convergence in male sterilization was visible mainly due to a decline in its use across the states which usually had a higher level of male sterilization. Except for Himachal Pradesh and Sikkim, the use of male sterilization across states came down to below 2 percent in 2015–2016. Apart from limiting methods, medium-acting reversible methods like IUD also observed a decline in many states. Only the North-Eastern states, Punjab, Gujarat, and Jammu and Kashmir, had higher use of IUD than the national average (Table A2).

Moreover, the use of short-span contraceptive methods, such as oral pills and condoms, has shown significant upward trends in many states. In particular, the uptake of oral pills was higher in the North-Eastern states, West Bengal, Odisha, and Jammu and Kashmir, whereas the use of condoms was higher in the states Delhi, Uttarakhand, Goa, Punjab, Uttar Pradesh, Himachal Pradesh, Jammu and Kashmir, and Haryana than the national average. Hence, Indian states showed diverging trends in terms of the use of condoms and pills, which can be attributed to state-specific variation in FP program intervention.

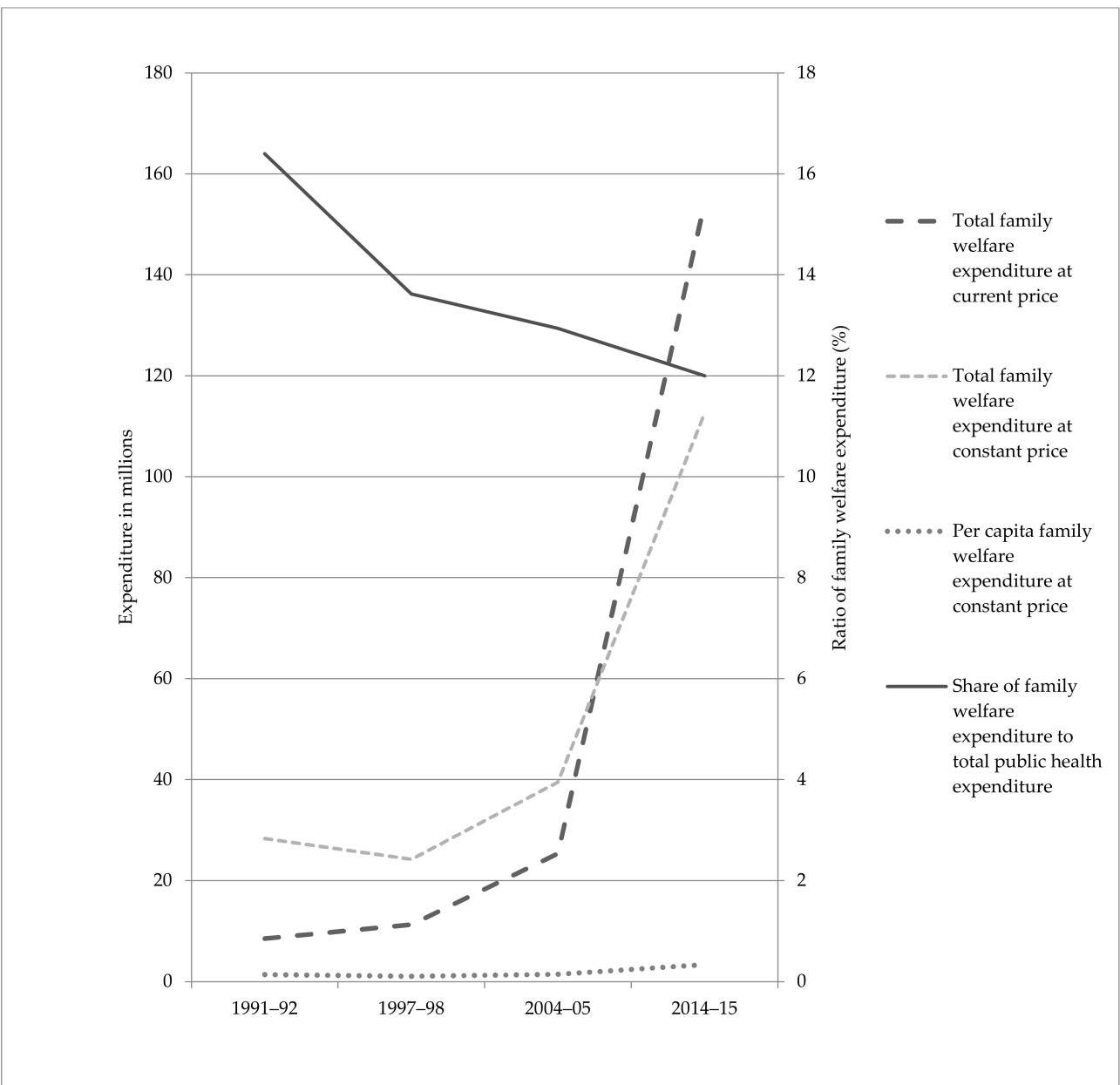

**Figure 1.** Family welfare expenditure at current and constant prices and per capita expenditure at constant prices in India, 1991–1992 to 2014–2015.Note: The total expenditure comprises the major states and union territories only. Expenditure in constant price is calculated considering the base year 2011–2012.

### 3.3. Sources of Contraceptive Methods

Table 1 provides trends and patterns of the public sector as the source of modern contraceptive methods and the PPR across the states by place of residence during 1992–1993 to 2015–2016. The results show that the source of modern contraceptives from the public sector reduced significantly in rural areas (a change of 13 percent between 1992–1993 and 2015–2016) compared to urban areas (9 percent between 1992–1993 and 2015–2016) at the aggregate level. The decline was also recorded across states and UTs. Between 1992–1993 and 2015–2016, the proportion of women using modern contraceptives from a public source decreased significantly in Arunachal Pradesh, Assam, Kerala, Meghalaya, Manipur, Mizoram, Tripura, Uttar Pradesh, and West Bengal. The decline was much higher in rural areas of Assam, Meghalaya, and Tripura, ranging as high as 50 percent. The urban areas of Nagaland, Orissa, and Rajasthan recorded a 46 percent decline at the same time.

**Table 1.** Public sector as the last source for current users and public-private source ratio for contraceptive access in India by states, 1992–1993 to 2015–2016.

| States | Total | | | | Urban | | | | Rural | | | | Public/Private Ratio (2015–2016) | | |
|---|---|---|---|---|---|---|---|---|---|---|---|---|---|---|---|
| | 1992–1993 | 1998–1999 | 2005–2006 | 2015–2016 | 1992–1993 | 1998–1999 | 2005–2006 | 2015–2016 | 1992–1993 | 1998–1999 | 2005–2006 | 2015–2016 | Total | Urban | Rural |
| All India | 79.0 | 76.0 | 69.8 | 68.7 | 62.4 | 60.1 | 55.6 | 56.7 | 87.0 | 83.2 | 77.6 | 75.5 | 2.2 | 1.3 | 3.1 |
| Andhra Pradesh | 78.1 | 78.5 | 76.6 | 74.7 | 61.8 | 64.9 | 67.2 | 64.7 | 85.6 | 83.4 | 81.0 | 78.9 | 2.9 | 1.8 | 3.7 |
| Arunachal Pradesh | 85.7 | 72.2 | 63.1 | 59.9 | 72.2 | 64.3 | 49.3 | 52.2 | 89.6 | 74.2 | 69.0 | 62.0 | 1.5 | 1.1 | 1.6 |
| Assam | 72.2 | 63.7 | 44.2 | 40.7 | 55.8 | 46.8 | 28.1 | 30.5 | 76.5 | 65.5 | 49.3 | 42.4 | 0.7 | 0.4 | 0.7 |
| Bihar | 76.1 | 76.9 | 53.4 | 63.1 | 55.0 | 56.0 | 39.5 | 51.3 | 83.7 | 80.9 | 56.9 | 65.5 | 1.7 | 1.0 | 1.9 |
| Chandigarh | — | — | — | 54.8 | — | — | — | 53.2 | — | — | — | 86.7 | 1.2 | 1.1 | 6.5 |
| Delhi | 45.2 | 51.9 | 43.9 | 51.8 | 45.2 | 50.8 | 42.0 | 51.8 | 46.2 | 63.7 | 65.7 | 60.7 | 1.1 | 1.0 | 1.5 |
| Gujarat | 75.5 | 72.0 | 70.6 | 70.0 | 63.6 | 52.2 | 53.8 | 53.7 | 82.3 | 86.7 | 83.5 | 82.0 | 2.4 | 1.2 | 4.6 |
| Goa | 72.0 | 68.3 | 58.9 | 50.5 | 62.7 | 62.8 | 51.9 | 50.8 | 80.7 | 72.9 | 69.3 | 48.9 | 1.0 | 1.0 | 0.9 |
| Himachal Pradesh | 90.6 | 91.7 | 84.6 | 81.0 | 75.2 | 68.4 | 61.3 | 66.2 | 92.6 | 94.2 | 87.2 | 82.4 | 4.3 | 2.0 | 4.7 |
| Haryana | 83.1 | 79.5 | 68.1 | 69.1 | 65.8 | 59.2 | 51.4 | 58.3 | 90.1 | 87.9 | 75.3 | 75.5 | 2.2 | 1.4 | 3.1 |
| Jharkhand | — | — | 57.4 | 65.5 | — | — | 47.4 | 50.9 | — | — | 63.7 | 71.3 | 1.9 | 1.0 | 2.5 |
| Jammu and Kashmir [a] | 81.1 | 68.5 | 57.9 | 65.5 | 62.8 | 60.7 | 52.3 | 60.3 | 86.2 | 72.0 | 61.2 | 68.7 | 1.9 | 1.5 | 2.2 |
| Karnataka | 83.4 | 85.3 | 82.1 | 83.3 | 68.9 | 70.7 | 69.0 | 73.4 | 90.9 | 93.3 | 89.8 | 89.6 | 5.0 | 2.8 | 8.8 |
| Kerala | 74.9 | 66.4 | 61.3 | 57.8 | 72.1 | 63.4 | 57.9 | 54.5 | 76.1 | 67.4 | 63.1 | 60.7 | 1.4 | 1.2 | 1.5 |
| Meghalaya | 68.2 | 47.3 | 42.5 | 42.9 | 58.5 | 33.5 | 40.7 | 44.1 | 73.5 | 61.8 | 44.1 | 42.6 | 0.8 | 0.8 | 0.7 |
| Maharashtra | 74.8 | 75.2 | 68.4 | 69.7 | 55.2 | 59.1 | 52.0 | 55.3 | 87.8 | 85.5 | 83.2 | 81.8 | 2.3 | 1.2 | 4.5 |
| Madhya Pradesh | 89.2 | 86.6 | 86.1 | 85.6 | 74.9 | 69.8 | 65.2 | 68.1 | 94.9 | 94.2 | 94.2 | 92.6 | 6.0 | 2.2 | 12.8 |
| Manipur | 82.3 | 76.9 | 51.4 | 47.6 | 80.4 | 69.7 | 43.4 | 43.2 | 83.7 | 81.7 | 55.4 | 50.4 | 0.9 | 0.8 | 1.0 |
| Mizoram | 90.6 | 81.0 | 83.9 | 67.9 | 87.4 | 72.8 | 80.5 | 59.9 | 94.0 | 92.8 | 88.6 | 79.1 | 2.1 | 1.5 | 3.9 |
| Nagaland | 70.7 | 57.7 | 47.2 | 59.7 | 68.9 | 59.5 | 40.9 | 52.6 | 71.6 | 56.8 | 51.4 | 64.6 | 1.5 | 1.1 | 1.8 |
| Orissa | 93.4 | 88.5 | 78.1 | 76.0 | 79.7 | 74.2 | 59.5 | 58.0 | 96.8 | 91.6 | 82.4 | 80.1 | 3.2 | 1.4 | 4.1 |
| Punjab | 77.1 | 64.3 | 61.4 | 63.7 | 62.7 | 40.3 | 43.5 | 52.1 | 83.1 | 75.2 | 70.7 | 71.2 | 1.8 | 1.1 | 2.5 |
| Rajasthan | 92.3 | 86.3 | 80.8 | 76.9 | 85.6 | 73.3 | 62.3 | 59.6 | 95.1 | 91.8 | 91.8 | 83.2 | 3.4 | 1.5 | 4.9 |
| Sikkim | — | 77.9 | 65.4 | 74.9 | — | 76.3 | 50.8 | 55.0 | — | 78.2 | 69.1 | 81.6 | 2.9 | 1.2 | 4.5 |
| Tamil Nadu | 78.0 | 73.6 | 72.0 | 77.2 | 63.3 | 65.0 | 65.7 | 70.8 | 85.8 | 78.9 | 77.4 | 84.0 | 3.4 | 2.4 | 5.3 |
| Tripura | 75.3 | 71.4 | 53.2 | 39.1 | 69.6 | 57.9 | 35.0 | 40.3 | 77.4 | 75.2 | 57.0 | 38.6 | 0.6 | 0.7 | 0.6 |
| Uttarakhand | — | — | 65.0 | 63.3 | — | — | 41.3 | 45.7 | — | — | 74.2 | 72.8 | 1.7 | 0.8 | 2.7 |
| Uttar Pradesh | 74.5 | 71.1 | 58.4 | 53.8 | 55.2 | 49.8 | 40.3 | 33.9 | 83.6 | 81.7 | 67.9 | 62.8 | 1.2 | 0.5 | 1.7 |
| West Bengal | 79.6 | 69.5 | 64.2 | 57.4 | 57.9 | 52.6 | 51.8 | 44.0 | 87.6 | 74.6 | 69.1 | 62.7 | 1.4 | 0.8 | 1.7 |

[a] Jammu region of Jammu and Kashmir for NFHS-1.

The estimates of PPR showed higher public sector dependency (i.e., the ratio of more than 1 indicates public sector dependency) in India and across the state. However, the reliance was higher in rural areas than in the urban areas. The PPR is 3.2 in rural areas compared to only 1.3 in urban areas in 2015–2016. Among all states, the PPR was highest in the rural areas of Madhya Pradesh (12.8), followed by Tamil Nadu (5.3), and Himachal Pradesh (4.7). Assam shows the least PPR in both rural and urban areas (0.4 in rural and 0.7 in urban areas).

Figure 2 presents the sources of popularly used modern contraceptive methods between 1992–1993 and 2015–2016. Evidence suggested that for long-acting and surgical methods like sterilization (both for male and female), more than 80 percent population used public sources. Even though public sources remained the most preferred source for sterilization, the study found a change in the source-mix for other contraceptive methods over time. It showed a decline in the public sector as the last source for obtaining sterilization and a slight increase in obtaining spacing methods for the recent period, 2015–2016.

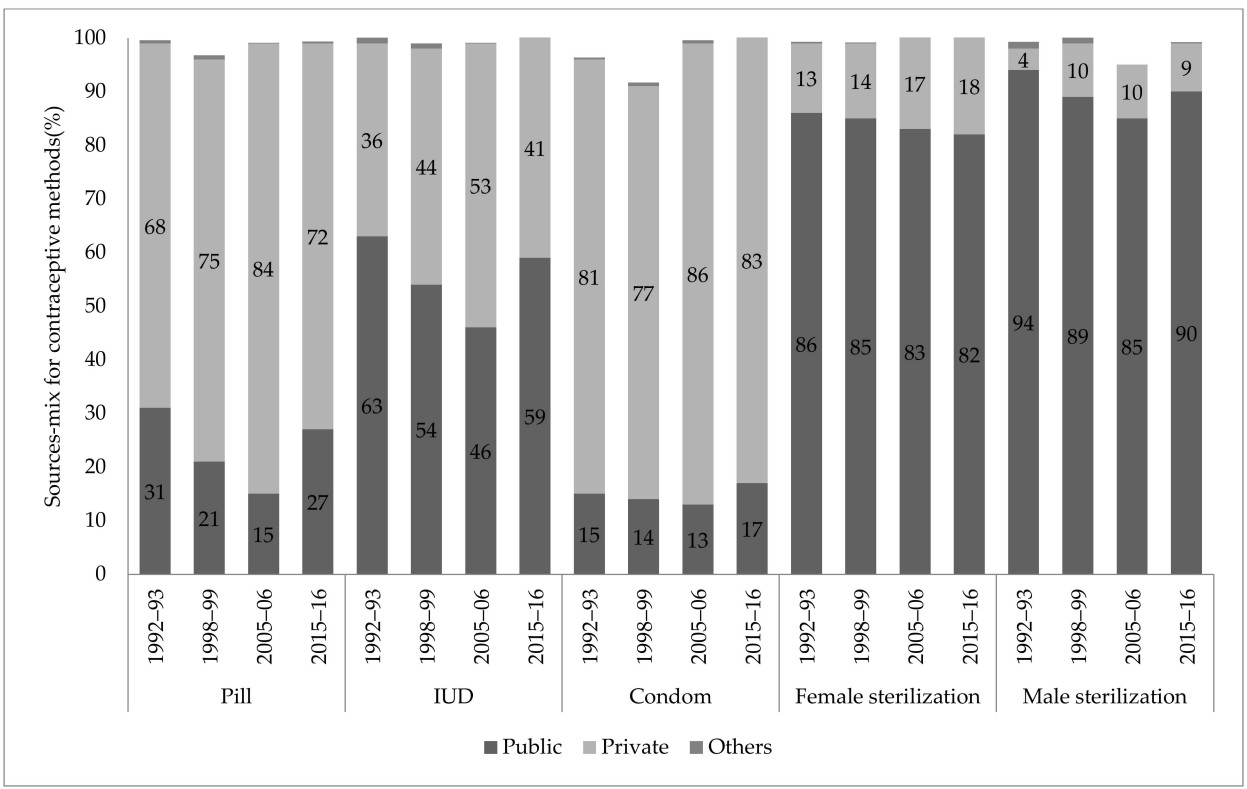

**Figure 2.** Source-mix for different contraceptive methods in India, 1992–1993 to 2015–2016.

In contrast, users mostly depend on private sources to obtain short to medium-acting spacing methods such as IUDs, pills, and condoms. During the entire period (1992–1993 to 2015–2016), private sources consisted of a large base to supply pills and condoms (more than 50 percent) in India. The recent data of 2015–2016 showed that the private sector supply of these two methods was 72.4 percent and 82.7 percent for pills and condoms, respectively. Among all short and medium-acting methods, only IUDs had a balanced source between public and private sectors.

Table 2 presents the pattern of modern methods of contraceptive use from public sources by women's socio-economic status. The results show that the use of public sources for any modern contraceptives including female sterilization varied according to the socio-economic status of current users. The contraceptive use from the public sector showed a decline among socio-economically privileged sections. For the year 2015–2016, women from the poorest and poorer wealth quintiles mostly accessed any modern contraception from the public sector (82 percent for any modern contraception and 92.5 percent for female sterilization). In the same year, the wealthiest counterparts were less likely to access contraceptive services from the public sector (45.1 percent for modern contraceptives and 61.8 percent for female sterilization). Moreover, the public source for FP access remained higher among rural women (75 percent for modern contraceptives and 87 percent for female sterilization) compared to their urban counterparts. A similar pattern is observed in the case of the use of any modern contraceptives by educational categories in 2015–2016. Among women with higher education, access to modern contraceptive methods (30.4 percent) and female sterilization (45.3 percent) from the public sector is lower than their less-educated counterparts. Even for a recent decade (2005–2006 to 2015–2016), trends in any modern contraceptives by source suggest a small rise in public sector provision for both the richest wealth quintile and higher educated groups.

### 3.4. Correlation of FWE and PPR With Use of Modern Contraceptive Methods

Figure 3 shows the bivariate relationship between the current use of modern contraceptive methods and the percentage share of FWE in PHE for the latest year, 2015–2016. The scatter plots indicated a moderate positive relationship between public funding and the prevalence of modern contraceptives among current users. Moreover, there is a slightly higher positive correlation between the share of FWE in PHE and female sterilization across states. But the share of FWE in PHE is either negatively associated or not related to the use of reversible methods and male sterilization.

**Table 2.** Percentage distribution of public source for modern contraceptive use and female sterilization by socioeconomic background characteristics in India, 1992–1993 to 2015–2016.

| Characteristic | Any Modern Contraceptives | | | | Female Sterilization | | | |
|---|---|---|---|---|---|---|---|---|
| | 1992–1993 | 1998–1999 | 2005–2006 | 2015–2016 | 1992–1993 | 1998–1999 | 2005–2006 | 2015–2016 |
| Wealth status | | | | | | | | |
| Poorest | 90.2 | 92.3 | 87.6 | 82.0 | 96.1 | 96.1 | 95.4 | 92.5 |
| Poorer | 89.6 | 88.7 | 82.6 | 78.1 | 94.9 | 93.0 | 91.3 | 91.2 |
| Middle | 88.3 | 83.3 | 79.3 | 75.9 | 92.4 | 89.4 | 88.7 | 87.3 |
| Richer | 83.1 | 71.7 | 68.2 | 67.6 | 87.0 | 80.8 | 80.4 | 79.5 |
| Richest | 68.3 | 48.3 | 45.1 | 47.4 | 71.8 | 63.7 | 63.0 | 61.8 |
| Place of residence | | | | | | | | |
| Urban | 62.4 | 60.1 | 55.6 | 56.7 | 78.9 | 74.5 | 73.6 | 71.9 |
| Rural | 89.0 | 83.2 | 77.6 | 75.5 | 92.1 | 89.4 | 87.5 | 87.0 |
| Education | | | | | | | | |
| No education | 90.1 | 88.9 | 83.4 | 83.6 | 93.9 | 92.6 | 90.6 | 90.8 |
| Primary | 80.9 | 79.4 | 76.4 | 76.6 | 87.2 | 86.3 | 86.2 | 87.3 |
| Secondary | 61.5 | 63.7 | 57.1 | 61.6 | 79.7 | 75.3 | 72.6 | 75.7 |
| Higher | 28.2 | 33.4 | 22.7 | 30.4 | 65.3 | 51.2 | 42.8 | 45.3 |

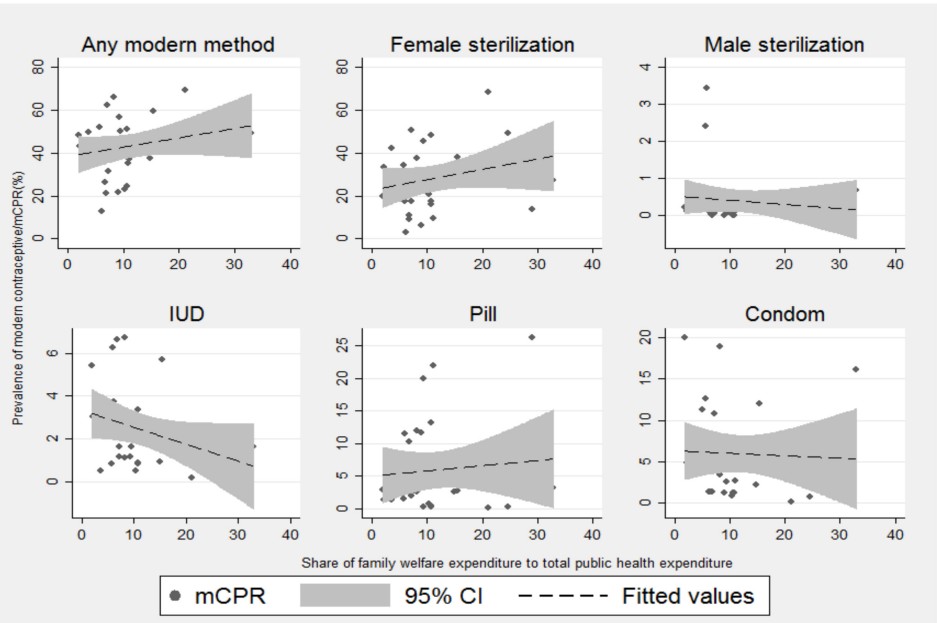

**Figure 3.** Correlation between the prevalence of modern contraceptive (mCPR) and percent share of family welfare expenditure in total public health expenditure in India, 2015–2016. Note: mCPR stands for the percentage of modern contraceptive prevalence rate.

A similar pattern is also observed in the case of the public-private ratio (PPR) and the use of different contraceptive methods. Overall prevalence of any modern contraceptives, especially female sterilization, is positively associated with PPR, indicating public healthcare facilities as the primary source for limiting methods (Figure 4). However, the PPR was negatively correlated with the spacing methods, especially the use of IUDs and pills.

*3.5. The Effect of FWE on FP Use: Results From Panel Data Regression Analyses*

Table 3 shows the results of the panel data regression analyses showing the effect of FWE on FP indicators. All the models were controlled for relevant socio-economic and demographic predictors. Multi-collinearity was considered while selecting co-variates for each model. The results from Model 1 shows the effect of FWE on the use of any modern contraceptive methods is negative and significant ($\beta = -4.886$, $p < 0.1$) after controlling for under-five mortality, antenatal care, child marriage, female

literacy, the sex ratio at birth, proportion SC/ST, and Muslim population. Among other predictors, under-five mortality (β = −0.087, *p* < 0.1) and proportion of SC/ST population (β = −0.161, *p* < 0.1) are significantly negatively associated with the use of modern methods, while other factors are not significantly associated with this outcome variable. Interestingly, the results in Model 2 show that the main predictor variable, the share of FWE in PHE (β = 0.076, *p* < 0.1) is not significantly associated with the use of modern methods. Among other factors, only child marriage (β = −0.351, *p* < 0.01) and per capita NSDP (β = −25.775, *p* < 0.01) are significantly associated with the use of modern methods among users.

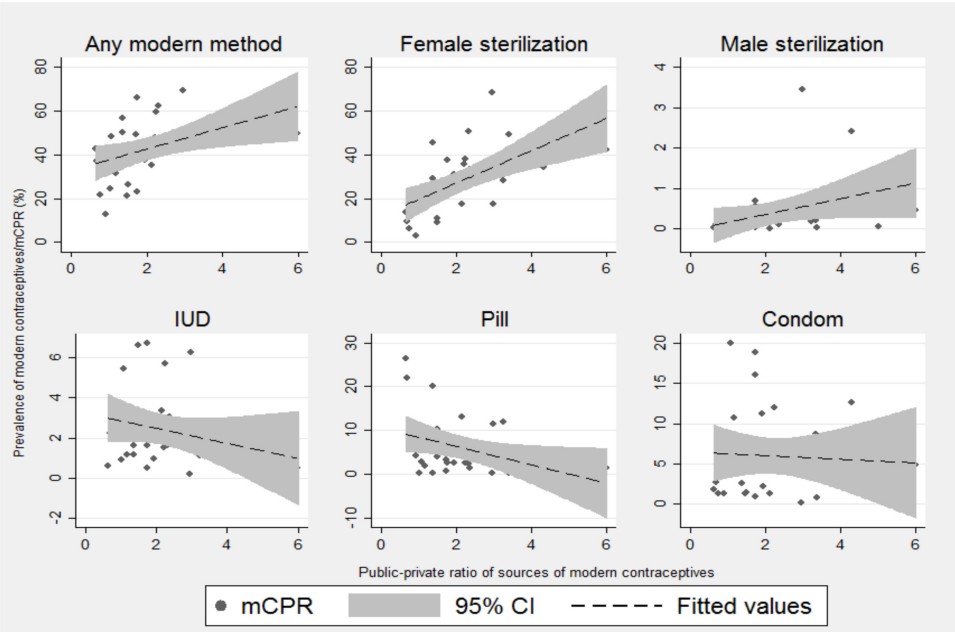

**Figure 4.** Correlation between public-private ratio (PPR) and prevalence of modern contraceptive in India, 2015–2016. Note: mCPR stands for the percentage of modern contraceptive prevalence rate.

Models 3 and 4 demonstrate the net effect of FWE and the ratio of FWE to PHE on the use of any modern spacing methods. The results in Model 3 show that per capita FWE is significantly negatively associated with the use of spacing methods (β= −2.446, *p* < 0.1). Among other factors, antenatal care (ANC) has shown a positive association with the use of spacing methods (β = 0.190, *p* < 0.01), meaning the states with higher use of spacing methods also have a higher level of ANC visits. The net effect of the ratio of FWE to PHE (β = 0.118, *p* < 0.01) on the use of spacing methods indicates no significant association, while the level of ANCs (β = 0.195, *p* < 0.01) shows a positive relationship with the use of any modern spacing methods.

*3.6. Factors Associated with the Public Sector as the Source of Modern Contraception: Results from Binary Logistic Regression Models*

The effects of socio-economic and demographic factors including types of contraceptive methods on the source of modern contraceptive methods are assessed using a binary logit regression model (Table 4). The results show that limiting methods have a strong association with the public sector compared to spacing methods. In particular, uptake of pills and condoms is significantly less from the public sector compared to other methods.

Moreover, there is significant regional variation in accessing the public sector as the source of the modern contraceptive method across the states of India. The state-like Haryana, Himachal Pradesh, Jammu and Kashmir, Madhya Pradesh, and Uttarakhand showed higher levels of access to the public sector for their contraceptive methods. However, states such as Bihar, Uttar Pradesh, Jharkhand, Kerala, and Gujarat showed lower levels of access to the public sector for contraceptive services. The use of contraceptive services from the public sector is significantly less among women from the richest wealth quintile and educated women compared to their poor and less educated counterparts. But in contrast, more working women accessed their contraceptive methods from public services than non-working women. Among other socio-demographic characteristics, access to

mass media, the current age of respondents, and age at marriage showed significant variations in using public sources to obtain contraception.

**Table 3.** Results of Fixed and Random effects model: Effect of public health spending on family planning indicators in India.

| Indicators | Any Modern Method | | | | Any Modern Method for Spacing | | | |
| --- | --- | --- | --- | --- | --- | --- | --- | --- |
| | Model1 (Re) | | Model 2 (Fe) | | Model 3 (fe) | | Model 4 (Fe) | |
| | $\beta$ | *p*-Value | B | *p*-Value | B | *p*-Value | $\beta$ | *p*-Value |
| Log of per capita FWE | −4.886 (2.544) | 0.055 | | | −2.446 (1.297) | 0.064 | | |
| Share of FWE in PHE | | | 0.076 (0.133) | 0.565 | | | 0.118 (0.079) | 0.139 |
| Under five mortality | −0.087 (0.057) | 0.126 | −0.040 (0.055) | 0.456 | −0.0006 (0.032) | 0.984 | 0.001 (0.033) | 0.979 |
| ANC (At least 4 visits) | 0.096 (0.080) | 0.229 | 0.087 (0.087) | 0.316 | 0.190 (0.051) | 0.000 | 0.195 (0.052) | 0.000 |
| Child marriages | −0.081 (0.089) | 0.358 | −0.351 (0.093) | 0.000 | 0.019 (0.053) | 0.721 | 0.022 (0.056) | 0.695 |
| Female literacy rate | 0.123 (0.103) | 0.229 | 0.081 (0.112) | 0.474 | 0.039 (0.063) | 0.534 | 0.035 (0.067) | 0.608 |
| Urban population | 0.043 (0.113) | 0.700 | 0.155 (0.220) | 0.483 | −0.093 (0.125) | 0.456 | −0.138 (0.131) | 0.298 |
| Sex ratio at Birth (SRB) | −0.007 (0.014) | 0.587 | 0.009 (0.013) | 0.480 | 0.012 (0.007) | 0.109 | 0.010 (0.008) | 0.194 |
| SC/ST population | −0.161 (0.082) | 0.048 | −0.080 (0.136) | 0.556 | 0.038 (0.077) | 0.615 | 0.034 (0.081) | 0.666 |
| Muslim population | −0.080 (0.126) | 0.522 | −0.108 (0.142) | 0.450 | 0.019 (0.080) | 0.805 | 0.021 (0.085) | 0.801 |
| Log of per capita NSDP | | | −25.775 (6.062) | 0.000 | | | −1.747 (3.611) | 0.630 |
| Constant | 492.311 (302.593) | 0.104 | −1175.57 (654.710) | 0.077 | −214.91 (269.31) | 0.428 | −445.584 (390.115) | 0.258 |
| N | 102 | | 102 | | 102 | | 102 | |
| sigma_u | 9.466 | | 14.252 | | 8.293 | | 8.752 | |
| sigma_e | 5.661 | | 5.156 | | 3.023 | | 3.072 | |
| rho | 0.736 | | 0.884 | | 0.882 | | 0.890 | |

Note: Fe: Fixed effect model, Re: Random effect model, Standard errors in parentheses.

**Table 4.** Results from binary logistic regression model: Odds of using public sector as a source of modern contraception over private sector by background characteristics, 2015–2016.

| Variables | OR | *p*-Value | (95% CI) |
| --- | --- | --- | --- |
| Type of contraceptive methods | | | |
| Female sterilization | Ref. | | |
| Pill | 0.05 | 0.000 | (0.05 to 0.06) |
| IUD | 0.44 | 0.000 | (0.41 to 0.46) |
| Condom | 0.04 | 0.000 | (0.04 to 0.04) |
| Other | 0.10 | 0.000 | (0.08 to 0.11) |

**Table 4.** *Cont.*

| Variables | OR | *p*-Value | (95% CI) |
|---|---|---|---|
| **State** | | | |
| Andhra Pradesh | 0.26 | 0.000 | (0.23 to 0.28) |
| North East | 0.39 | 0.000 | (0.36 to 0.42) |
| Bihar | 0.12 | 0.000 | (0.11 to 0.13) |
| Chhattisgarh | 0.86 | 0.002 | (0.77 to 0.95) |
| Gujarat | 0.46 | 0.000 | (0.42 to 0.51) |
| Haryana | 1.00 | 0.927 | (0.91 to 1.08) |
| Himachal Pradesh | 2.34 | 0.000 | (2.06 to 2.65) |
| Jammu and Kashmir | 1.18 | 0.002 | (1.06 to 1.30) |
| Jharkhand | 0.18 | 0.000 | (0.16 to 0.19) |
| Karnataka | 0.57 | 0.000 | (0.52 to 0.62) |
| Kerala | 0.28 | 0.000 | (0.25 to 0.30) |
| Madhya Pradesh | 1.18 | 0.000 | (1.09 to 1.29) |
| Maharashtra | 0.52 | 0.000 | (0.48 to 0.57) |
| Delhi | 0.86 | 0.044 | (0.75 to 1.00) |
| Odisha | 0.97 | 0.467 | (0.88 to 1.06) |
| Punjab | Ref. | | |
| Rajasthan | 0.83 | 0.000 | (0.77 to 0.91) |
| Tamil Nadu | 0.39 | 0.000 | (0.36 to 0.42) |
| Uttar Pradesh | 0.35 | 0.000 | (0.32 to 0.38) |
| Uttarakhand | 1.02 | 0.769 | (0.92 to 1.12) |
| West Bengal | 0.46 | 0.000 | (0.42 to 0.51) |
| Telangana | 0.19 | 0.000 | (0.17 to 0.21) |
| Goa and UTs | 0.75 | 0.000 | (0.67 to 0.84) |
| **Age of respondent (years)** | | | |
| 15–21 | Ref. | | |
| 22–34 | 1.07 | 0.058 | (1.00 to 1.15) |
| 35+ | 1.11 | 0.007 | (1.03 to 1.19) |
| **Age at marriage (years)** | | | |
| <14 | Ref. | | |
| 15–19 | 0.96 | 0.025 | (0.92 to 0.99) |
| 20–24 | 0.87 | 0.000 | (0.84 to 0.91) |
| 25–29 | 0.92 | 0.012 | (0.86 to 0.98) |
| 30+ | 0.98 | 0.730 | (0.85 to 1.12) |
| Not reported | 1.33 | 0.000 | (1.23 to 1.44) |
| **Place of residence** | | | |
| Urban | Ref. | | |
| Rural | 1.32 | 0.000 | (1.29 to 1.36) |
| **Religion** | | | |
| Hindu | Ref. | | |
| Muslim | 0.70 | 0.000 | (0.67 to 0.73) |
| Other | 1.06 | 0.020 | (1.01 to 1.11) |
| **Caste** | | | |
| Others | Ref. | | |
| SC | 1.35 | 0.000 | (1.30 to 1.39) |
| ST | 1.88 | 0.000 | (1.79 to 197) |
| OBC | 0.95 | 0.117 | (0.89 to 1.01) |
| **Wealth Quintiles** | | | |
| Poorest | Ref. | | |
| Poorer | 0.80 | 0.000 | (0.77 to 0.84) |
| Middle | 0.68 | 0.000 | (0.64 to 0.71) |
| Richer | 0.49 | 0.000 | (0.47 to 0.52) |
| Richest | 0.27 | 0.000 | (0.26 to 0.29) |

**Table 4.** *Cont.*

| Variables | OR | *p*-Value | (95% CI) |
|---|---|---|---|
| Respondent's education | | | |
| Illiterate | Ref. | | |
| Primary | 0.88 | 0.000 | (0.85 to 0.92) |
| Secondary | 0.68 | 0.000 | (0.66 to 0.70) |
| Higher | 0.37 | 0.000 | (0.35 to 0.39) |
| Occupation | | | |
| Not working | Ref. | | |
| Working | 1.23 | 0.000 | (1.15 to 1.32) |
| Don't know/ Not reported | 1.08 | 0.000 | (1.05 to 1.12) |
| Exposure to mass media | | | |
| No | Ref. | | |
| Partial | 0.98 | 0.160 | (0.95 to 1.01) |
| Full | 1.05 | 0.041 | (1.00 to 1.10) |
| Constant | 20.62 | 0.000 | (18.25 to 23.30) |
| N | | 228799 | |
| Chi-square test | | 96241.23 | |
| P-value | | 0.000 | |

Note: OR, odds ratio; CI, confidence interval.

## 4. Discussion

FP is still an unfinished agenda, as its impact could reach beyond the health benefits by meeting a better and sustainable future for all [2,49,50]. In this context, the present study makes a critical attempt by focusing on India's FP services and their relation to the change in FWE in the last two decades. The study also synthesized its intriguing findings using existing literature. The study put forth four findings. First, the findings advance that contraceptive use is not consistent with the amount of overall funding allocated for FWE programs. Interpretation of this finding in light of previous studies also suggests that the scale of rise in FWE is not uniform across all portfolios of the FP program. Furthermore, the inconsistent relationship between the increase in FWE and the rise in contraceptive use may be explained by the disproportionate allocation of funds to the core FP program within the FWE. The bulk of the funding is diverted to maternal and child health care programs rather than to the core FP activities as a part of the integration of FP and RCH programs into the umbrella of the family welfare programs [23,28,51]. On the other hand, some of the previous studies also suggest that as a result of the lack of significant rise in budget sanctions for family welfare programs, out-of-pocket payment for FP services is high. Out of pocket payments for female sterilizations are above 70 percent [26,34].

Second, the assessment of contraceptive method-mix reflects the skewed distribution towards female sterilization irrespective of regions of India. As pointed out earlier, although method-mix was analyzed in several previous studies, e.g., [37,40], its relationship with FWE is not explored. Skewness in the contraceptive method-mix in India can be explained through its close association with FWE and also through the lens of state policies, i.e., incentives in the FP programs since the 1950s. The incentives, particularly for female sterilization, were initiated to combat the increasing size of the population. Although modern spacing methods have been encouraged since the 1980s [29], the incentives for female sterilization continue to attract a significant share of individuals, especially in South Indian states [43,52]. On the contrary, male sterilization has sharply declined because of the backlash received on coercive approaches during the post-emergency period. Nonetheless, the reliance on sterilization is primarily dependent upon the user's socio-economic status and state-specific policies [53,54].

The medium-term contraceptive method, i.e., the use of IUDs, remains low even after the government-initiated programs such as the Postpartum Intrauterine Device (PPIUD) scheme. It indicates the failure of operational management and infrastructure, side effects, and health concerns among users [55–57]. However, the results show an increase in short-acting methods such as condoms and oral pills during the period. The use of pills and condoms from public sources is even slightly higher among poorer and poorest sections, and scheduled tribes and castes than their counterparts.

It is perhaps because of their close interaction with frontline health workers. Additionally, previous studies suggest that the higher use of IUD among scheduled castes and tribes may be attributable to less restriction on women accessing the method [58]. The government schemes of home delivery with free or minimal cost of services by frontline health workers (FHWs) and television and radio advertisements have also contributed towards gaining popularity for pills and condoms [59].

Third, our finding that the spacing methods are significantly varied across the states of India is in tune with previous studies [40,60]. Further, there is no systematic association between the use of spacing methods and the level of socio-economic development of states. In other words, irrespective of the background characteristics of the sample or state, lower birth interval and lower use of spacing methods are the major challenges for raising the overall contraception use in India.

Fourth, this study advances that the sources of modern contraception present a clear public-private divide across methods. Although the public sector as a source of limiting methods is decreasing, it remains the dominant sector for accessing FP for socio-economically poor individuals. Until now, the dependency on public sources is assumed to be a result of over-subsidization that puts up barriers to the promotion of private facilities at a low cost. Thus, so far, more dependence on private sources concentrated among the richest wealth quintile, urban residents, and users of spacing methods. Even in socio-economically backward states, private providers are the dominant source of obtaining spacing methods.

Finally, the study suggests that socioeconomic status has been an important factor for individuals to choose their source and type of contraception methods irrespective of region of India. In particular, due to financial constraints, individuals perhaps go to the public sector for sterilization after completing their childbearing because the spacing methods in the private sector are not cost-effective [26,60]. While previous studies suggest that the involvement of the private sector in national FP programs could expand the market, reducing economic disparity with subsidized services, it could also raise contraceptive usage in the future [61,62]. While another previous study suggested that the private sector in India can complement the public sector for FP services, the road map for bringing together these two sectors remains a challenge [41]. At the outset, considering the current trajectories in the source of contraceptive methods, it can be inferred that the pace of increase in accessing the private sector is lower than the pace at which the public sector access is decreasing. Evidence suggests that in some states (e.g., Uttar Pradesh and Bihar), the share of the public sector is quite low as compared to the national averages. Drawing inferences from this, we can submit that decline in the public sector and failure of the private sector to emerge as a cost-effective replacement might be one of the key reasons for stalling or decline in contraceptive use in India. In particular, the individuals who belong to socio-economic backward communities face the major hurdles of accessing contraceptive services from the public sector.

## 5. Conclusions

In conclusion, we suggest that access to the public sector for contraceptive use is a crucial step for universal access to FP; thus, this study calls for necessary policy actions. To bring FP programs and services on track, the Government of India has committed to investing 3 US billion dollars in reaching an additional 48 million women and girls who have unmet needs for FP by 2020 (Vision FP2020). Nevertheless, a decline in contraception prevalence, continued disparity in modern contraception use, the sluggish decline in the unmet needs for FP, and the rise in contraceptive discontinuation raise concerns over policy issues. The country needs to tackle the policy challenges ahead. As suggested previously and in this study, the lack of sufficient expansion of the private sector and a decline in core FWE relative to total PHE led to a decreasing overall modern contraceptive prevalence rate (mCPR) [63]. In particular, a focus on the reasons for the decrease in the use of IUDs and inadequate rise in pills, condoms, and other spacing methods should be given. Therefore, addressing the demands of disadvantaged groups strong public programs is critical. It has been projected that India will enjoy an additional per capita income of 13 percent during 2026–2031 if policies can provide an increase in investment and a multi-sectoral supply approach to meet existing demands [26,64,65]. Thus, this study identified a need for revitalizing the FP program to promote spacing methods in relatively lower-performing states.

**Author Contributions:** Conceptualization, S.M., S.G., and M.J.R.; methodology, S.M. and M.J.R.; software, S.M.; validation, M.J.R. and S.G.; formal analysis, S.M.; investigation, S.M.; resources, P.N.; data curation, S.M.; writing—original draft preparation, S.M. and M.J.R.; writing—review and editing, S.G., M.J.R., A.G., N.D., P.N. and R.V.; visualization, S.M.; supervision, S.G.; project

administration, S.G., A.G., N.D., P.N. and R.V. All authors have read and agreed to the published version of the manuscript."

**Funding:** This research was funded by the Bill & Melinda Gates Foundation, India country Office, New Delhi; Grant Number: OPP1142874.

**Institutional Review Board Statement:** Not applicable.

**Informed Consent Statement:** Not applicable.

**Data Availability Statement:** The data used in this study is drawn from the openly accessed data Demographic and Health Survey which is available from this url: https://dhsprogram.com/data/available-datasets.cfm (Accessed on 11 January 2020).

**Conflicts of Interest:** The authors declare no conflict of interest.

**Appendix A**

**Table A1.** State-wise share of family welfare expenditure in total public healthcare expenditure, 1995–1996 to 2014–2015.

| State | Share of Family Welfare in Total Public Healthcare Expenditure | | | |
|---|---|---|---|---|
| | 1995–1996 | 1997–1998 | 2004–2005 | 2014–2015 |
| India | 19.42 | 16.37 | 13.49 | 15.98 |
| Andhra Pradesh | 21.29 | 19.49 | 21.22 | 21.07 |
| Arunachal Pradesh | 3.19 | 4.60 | 4.6 | 6.68 |
| Assam | 16.69 | 12.78 | 17.11 | 11.04 |
| Bihar | 48.68 | 25.61 | 19.41 | 10.34 |
| Delhi | 6.67 | 3.65 | 1.26 | 1.93 |
| Goa | 2.6 | 3.66 | 2.77 | 2.18 |
| Gujarat | 16.19 | 15.30 | 14.01 | 10.7 |
| Haryana | 19.64 | 15.91 | 15.9 | 5.69 |
| Himachal Pradesh | 14.93 | 12.02 | 11.77 | 15.45 |
| Jammu & Kashmir | - | 8.72 | 9.28 | 14.88 |
| Jharkhand | - | - | - | 4.99 |
| Karnataka | 18.42 | 18.07 | 14.05 | 10.69 |
| Kerala | 14.7 | 14.13 | 14.06 | 9.38 |
| Madhya Pradesh | 17.24 | 15.42 | 12.5 | 8.99 |
| Maharashtra | 14.52 | 10.76 | 8.06 | 7.09 |
| Manipur | 15.62 | 14.09 | 14.85 | 3.64 |
| Meghalaya | 11.26 | 11.23 | 13.28 | 6.16 |
| Mizoram | 7.44 | 6.99 | 9.42 | 10.82 |
| Nagaland | - | 5.69 | 8.22 | 6.82 |
| Odisha | 23.11 | 20.08 | 20.53 | 8.18 |
| Punjab | 14.68 | 9.52 | 8.02 | 8.22 |
| Rajasthan | 20.22 | 20.97 | 18.77 | 31.27 |
| Sikkim | 12.81 | 12.32 | 11 | 5.89 |
| Tamil Nadu | 17.84 | 16.36 | 16.42 | 24.63 |
| Telangana | - | - | - | 26 |
| Tripura | 20.24 | 23.13 | 23.08 | 29.09 |
| Uttar Pradesh | 21.16 | 21.45 | 17.09 | 32.87 |
| Uttarakhand | - | - | - | 7.22 |
| West Bengal | 14.44 | 12.20 | 13.07 | 9.26 |

Source: Estimated by authors based on National Health account estimates for India, 2017, RBI's annual stud y on state finances.

Table A2. State-wise contraceptive method-mix in India, 1992–1993 to 2015–2016.

**Panel A**

| States | Female Sterilization | | | | Male Sterilization | | | | IUD | | | |
|---|---|---|---|---|---|---|---|---|---|---|---|---|
| | 1992-93 | 1998-99 | 2005-06 | 2015-16 | 1992-93 | 1998-99 | 2005-06 | 2015-16 | 1992-93 | 1998-99 | 2005-06 | 2015-16 |
| All India | 67.4 | 70.8 | 66.3 | 67.2 | 8.5 | 3.9 | 1.8 | 0.5 | 4.6 | 3.4 | 3.2 | 2.9 |
| Andhra Pradesh | 81.1 | 88.4 | 92.9 | 98.2 | 14.1 | 7.3 | 4.4 | 0.8 | 1.2 | 1.1 | 0.7 | 0.3 |
| Arunachal Pradesh | 43.6 | 58.1 | 52.1 | 35.4 | 1.5 | 0.3 | 0.2 | 0.1 | 19.3 | 11.8 | 8.3 | 10.8 |
| Assam | 28.3 | 36.3 | 23 | 18.2 | 5.5 | 2.3 | 0.4 | 0.2 | 2.1 | 4.3 | 2.3 | 4.2 |
| Bihar | 74.9 | 78.2 | 69.8 | 86 | 5.8 | 4 | 1.8 | 0.2 | 2.3 | 2.2 | 1.8 | 2.1 |
| Delhi | 33.2 | 41.2 | 34.4 | 36.2 | 5.4 | 3.7 | 1.2 | 0.4 | 13 | 9.7 | 7.5 | 9.9 |
| Gujarat | 76.1 | 72.8 | 64.4 | 71.6 | 7.2 | 3.9 | 0.9 | 0.2 | 6.1 | 5.2 | 6.8 | 6.5 |
| Goa | 61.8 | 58.5 | 53.5 | 61.8 | 2.1 | 0.8 | 0.2 | 0 | 5.7 | 4 | 4.8 | 3.5 |
| Himachal Pradesh | 55.9 | 66.5 | 67.5 | 60.6 | 22.5 | 10.8 | 8.7 | 4.2 | 4.6 | 3 | 2.1 | 1.5 |
| Haryana | 59.9 | 62 | 60.3 | 59.7 | 10.1 | 3.4 | 1.1 | 0.9 | 6.4 | 5.7 | 7.4 | 8.9 |
| Jharkhand | – | – | 65.5 | 77 | – | – | 1.1 | 0.5 | – | – | 1.7 | 2.4 |
| Jammu & Kashmir | 51.2 | 57 | 50 | 42.5 | 8.9 | 5.5 | 4.9 | 0.7 | 5.6 | 6 | 5.1 | 5 |
| Karnataka | 83.5 | 88.3 | 90.3 | 93.8 | 3.1 | 1.1 | 0.3 | 0.1 | 6.5 | 4.7 | 4.9 | 1.6 |
| Kerala | 66 | 76.2 | 71 | 86.2 | 10.3 | 3.9 | 1.5 | 0.1 | 4.3 | 2.5 | 3.4 | 3.1 |
| Meghalaya | 45.4 | 32.1 | 39.1 | 25.5 | 2.9 | 0 | 0.4 | 0 | 10.6 | 16.3 | 6.2 | 8.7 |
| Maharashtra | 74.4 | 79.5 | 76.4 | 78.3 | 11.5 | 6.1 | 3.1 | 0.7 | 4.7 | 3.2 | 4.6 | 2.5 |
| Madhya Pradesh | 72.3 | 80.6 | 79.2 | 82.2 | 14.1 | 5.1 | 2.3 | 0.9 | 3.1 | 1.8 | 1.3 | 1 |
| Manipur | 31.2 | 37.2 | 16.6 | 13.2 | 8.4 | 3 | 1 | 0.4 | 19.3 | 17.6 | 11.1 | 15.8 |
| Mizoram | 82.8 | 78.4 | 71.6 | 49.3 | 0.2 | 0.2 | 0 | 0 | 9.5 | 9.3 | 8.2 | 9.5 |
| Nagaland | 48.9 | 40.6 | 33.3 | 34.1 | 0.8 | 0 | 0 | 0 | 15.8 | 25.3 | 17.5 | 25.1 |
| Orissa | 77.8 | 72.3 | 65.3 | 49.3 | 9.3 | 3.7 | 2 | 0.3 | 4.2 | 1.7 | 1.2 | 2 |
| Punjab | 53.7 | 43.9 | 48.7 | 49.5 | 4.2 | 2.4 | 1.9 | 0.7 | 10.7 | 9.2 | 8.7 | 8.9 |
| Rajasthan | 79.7 | 76.4 | 72.5 | 68.2 | 7.4 | 3.6 | 1.7 | 0.4 | 3.9 | 2.9 | 3.4 | 2.1 |
| Sikkim | – | 41.5 | 36.8 | 37.6 | – | 4.4 | 7.8 | 7.4 | – | 10.4 | 5.4 | 13.4 |
| Tamil Nadu | 75.4 | 86.7 | 89.6 | 92.9 | 3.9 | 1.5 | 0.7 | 0 | 7.1 | 4.7 | 3.4 | 3.5 |
| Tripura | 29.9 | 47 | 26.7 | 21.7 | 4.3 | 1.1 | 0.8 | 0 | 2.7 | 3.5 | 1.5 | 0.9 |
| Uttarakhand | – | – | 54.1 | 51.3 | – | – | 3 | 1.3 | – | – | 2.5 | 3.1 |
| Uttar Pradesh | 59 | 53.2 | 39.7 | 38.1 | 6.9 | 2.4 | 0.5 | 0.1 | 5.8 | 3.5 | 3.2 | 2.6 |
| West Bengal | 45.9 | 47.9 | 45.2 | 41.3 | 7.5 | 2.8 | 1 | 0.1 | 2.2 | 2.1 | 0.8 | 1.7 |

**Panel B**

| States | Pill | | | | Condom | | | | Traditional methods | | | |
|---|---|---|---|---|---|---|---|---|---|---|---|---|
| | 1992-93 | 1998-99 | 2005-06 | 2015-16 | 1992-93 | 1998-99 | 2005-06 | 2015-16 | 1992-93 | 1998-99 | 2005-06 | 2015-16 |
| All India | 2.9 | 4.4 | 5.5 | 7.6 | 6 | 6.3 | 9.4 | 10.5 | 9.4 | 9.8 | 10.1 | 8.4 |
| Andhra Pradesh | 1 | 0.9 | 0.4 | 0.2 | 1.6 | 1.2 | 0.7 | 0.2 | 0.5 | 0.8 | 0.8 | 0.1 |
| Arunachal Pradesh | 13.7 | 20.5 | 19.2 | 32.3 | 3 | 1.9 | 6.7 | 4.5 | 17.3 | 6.2 | 9 | 7.9 |
| Assam | 6.4 | 14.6 | 18.2 | 42 | 4 | 4.1 | 4.2 | 5.2 | 46.7 | 33.7 | 35 | 22.9 |
| Bihar | 4.7 | 4 | 3.8 | 3.2 | 5.6 | 2.9 | 6.7 | 4 | 5.8 | 6.2 | 13.4 | 2.8 |
| Delhi | 4.8 | 6.3 | 6.7 | 5.3 | 34 | 27.4 | 34.8 | 36.5 | 8.9 | 10.7 | 9.8 | 6.8 |
| Gujarat | 2.1 | 2.6 | 3.9 | 2.9 | 3.6 | 5.9 | 9 | 10.4 | 4.6 | 8.9 | 11.5 | 4.5 |

**Table A2.** *Cont.*

**Panel B**

| States | Pill | | | | Condom | | | | Traditional methods | | | |
|---|---|---|---|---|---|---|---|---|---|---|---|---|
| | 1992-93 | 1998-99 | 2005-06 | 2015-16 | 1992-93 | 1998-99 | 2005-06 | 2015-16 | 1992-93 | 1998-99 | 2005-06 | 2015-16 |
| Goa | 1.5 | 1.9 | 3.1 | 1.3 | 8.2 | 10.3 | 16.2 | 26.9 | 19.1 | 20.3 | 13.8 | 4.7 |
| Himachal Pradesh | 0.9 | 1.9 | 3.9 | 2.7 | 9.1 | 7.4 | 16.1 | 22.2 | 6.3 | 9.5 | 1.7 | 6.2 |
| Haryana | 2.5 | 3.3 | 4.4 | 4.2 | 10.4 | 10.8 | 18.6 | 18.8 | 10.2 | 13.7 | 5.8 | 5 |
| Jharkhand | – | – | 10.6 | 6.4 | – | – | 7.8 | 5.4 | – | – | 10.7 | 5.7 |
| Jammu & Kashmir [a] | 2.7 | 6.7 | 8.9 | 10.9 | 11.9 | 9.8 | 15.4 | 19.7 | 18.7 | 13.2 | 9.3 | 13.3 |
| Karnataka | 0.9 | 1 | 1.7 | 0.9 | 2.4 | 1.8 | 3.3 | 2.5 | 3.2 | 2.7 | 1.1 | 0.4 |
| Kerala | 0.7 | 0.7 | 0.7 | 0.4 | 4.6 | 4.8 | 8.5 | 4.8 | 12.8 | 11.1 | 11.4 | 4.1 |
| Meghalaya | 11.6 | 22.1 | 20.2 | 47.9 | 2.4 | 6.2 | 10.3 | 5.3 | 7.6 | 18.1 | 13.2 | 5.3 |
| Maharashtra | 2.5 | 2.9 | 3.7 | 3.6 | 4.7 | 6.6 | 9.6 | 11 | 1.9 | 1.5 | 2.1 | 2.1 |
| Madhya Pradesh | 1.8 | 2.2 | 3 | 2.6 | 6 | 6.5 | 8.8 | 9.5 | 1.9 | 3 | 4.2 | 2.2 |
| Manipur | 6.8 | 5.7 | 10.9 | 17.9 | 3.5 | 3.4 | 8.6 | 5.7 | 28.9 | 29.9 | 31 | 29.8 |
| Mizoram | 4.7 | 9.3 | 17.7 | 37.4 | 1.2 | 1.6 | 2.8 | 3.6 | 1.4 | 1 | 0.3 | 0.1 |
| Nagaland | 16.5 | 8.4 | 15.8 | 15.1 | 16.5 | 5.8 | 9.4 | 5 | – | 18.2 | 16.1 | 12.1 |
| Orissa | 2.4 | 6.4 | 13.8 | 21 | 1.7 | 1.9 | 6.3 | 6 | 3.1 | 11.3 | 7.9 | 16 |
| Punjab | 3.7 | 4.6 | 4.6 | 3.3 | 15.2 | 20.6 | 24.5 | 24.9 | 11.9 | 17.9 | 8.1 | 8.7 |
| Rajasthan | 1.6 | 3.8 | 4.2 | 4 | 4.6 | 7.7 | 12.3 | 14.5 | 2.4 | 4.6 | 3.9 | 8 |
| Sikkim | – | 17.8 | 22.2 | 24.7 | – | 2.8 | 7.3 | 11.1 | – | 21.7 | 10.3 | 0.9 |
| Tamil Nadu | 1.1 | 0.6 | 0.3 | 0.4 | 3.2 | 2.9 | 3.7 | 1.6 | 7.4 | 3.1 | 1.7 | 0.9 |
| Tripura | 11.4 | 24.3 | 33.4 | 41.1 | 2.9 | 2.6 | 5.3 | 2.9 | 43.8 | 19.8 | 22.4 | 26.5 |
| Uttarakhand | – | – | 7.1 | 6.1 | – | – | 26.5 | 30.1 | – | – | 4.6 | 4.7 |
| Uttar Pradesh | 5 | 4.4 | 3.9 | 4.2 | 16.2 | 14.8 | 20 | 23.7 | 5.5 | 19.5 | 21.5 | 22.2 |
| West Bengal | 6.2 | 13.7 | 16.4 | 28.2 | 3.2 | 4.4 | 6.3 | 8.4 | 31.4 | 25.9 | 23.9 | 13.9 |

[a] Jammu region of Jammu and Kashmir for NFHS-1. Source: Estimated by authors.

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
