# Peer review of "Family Welfare Expenditure, Contraceptive Use, Sources and Method-Mix in India"

_sustainability, doi:10.3390/su13179562_

Round 1

Reviewer 1 Report

The manuscript addresses a timely issue that is common in middle-income developing countries, where there is a tendency to shift from publicly funded family planning services to privately provision of FP, either by insurance coverage or out-of-pocket expenditure. In this process, it is not clear how much and which groups of population will be affected if the private FP services cannot be affordable and accessible. This manuscript has nicely used macro time series data and micro data to investigate the trends of contraceptive use during a shift from public to private provision of FP services in India, and examines the factors associated with the use of FP in the public sector. I would suggest the following comments toward improvement of the manuscript.  

  • The manuscript has targeted three objectives, which need to be addressed by two to three separate papers separately and in depth each one. There is rooms for exploring in depth each one of these objective, rather than broadly look at them in one paper. (If possible), I would suggest the authors to break down the manuscript into at least two papers, one addresses the first two macro-level objectives, and the other one examines deeply the third micro-level objective, stated on pages 2-3.
  • The description of the data are not adequate. What are the unit of analysis for macro data? It remained unknown to me until I reached to Results.
  • Figure 1: The legends for “Total family welfare expenditure at current price” is duplicated for two lines.
  • Line 297: revise “from 6 to 3”. It should be 3 to 6, as the trend is increasing.
  • Table 4: selecting “other” as Ref. for Cast covariate is not a right one. It does not make sense to compare SC with “Others”, which is not known, for example. Can you specify “others”?

Author Response

  • Comment 1: There is rooms for exploring in depth each one of these objective, rather than broadly look at them in one paper. (If possible), I would suggest the authors to break down the manuscript into at least two papers, one addresses the first two macro-level objectives, and the other one examines deeply the third micro-level objective, stated on pages 2-3.

Response: Thank you for this suggestion. It would have been interesting to explore the objectives in two separate papers. But, it seems slightly out of scope with the present submission. Often these issues have been discussed separately; however, we thought to bring all three issues together to make a comprehensive narrative for a policy perspective.

  • Comment 2: The description of the data are not adequate. What are the unit of analysis for macro data? It remained unknown to me until I reached to Results.

Response: Thank you for pointing this out. The unit of analysis for macro data used in our analysis is the geographical units in India,i.e. states and union territories. We have collated the macro-level data for 30 states and union territories. We have incorporated your suggestion and described the unit of analysis for both macro and microdata in the 'Data sources' section (Page 3, line 114-116).

  • Comment 3: Figure 1: The legends for “Total family welfare expenditure at current price” is duplicated for two lines.

Response: We apologies for this error. The figure (Figure 1) has been modified with the recommended corrections in the revised manuscript.

  • Comment 4: Line 297: revise “from 6 to 3”. It should be 3 to 6, as the trend is increasing.

Response: As suggested by the reviewer, we have modified the sentence in the revised manuscript accordingly.

  • Comment 5: Table 4: selecting “other” as Ref. for Cast covariate is not the right one. It does not make sense to compare SC with “Others”, which is not known, for example. Can you specify “others”?

Response: While we appreciate the reviewer’s feedback, we would like to clarify this comment. Here “other category” is not a miscellaneous group, rather a well-defined category in the Indian Caste system. In the "other" category of the "Caste" variable, we included populations associated with other than scheduled caste, scheduled tribe, or other backward castes. This section of the population generally has a superior position and is socio-economically better-off in the Indian society compared to the other three categories. There is clearly articulated evidence of persistent discrimination against the scheduled caste, scheduled tribe, and other backward castes. Therefore, by taking the "other" category as a reference, we attempted to identify if these three caste groups are still the under-served section and, if so, what is the magnitude of differences in the outcome variable (here use of public sector as a source of modern contraception) among the caste groups.

Reviewer 2 Report

Reviewer’s comment:

I think this is a good paper. Introduction is adequate and the goals are clearly stated. The methods and results are well presented. However, there are few sentences and  use of words  that are a bit unclear and could be improved.  I think this work will need a minor to moderate editing.

For example, on line 20 in the abstract, it will be helpful to substitute the word “rise” with the word “increase”.  

Also, on page 2, line 60, authors stated:  “ However, limiting is just one of the several goals...”  What do they meant by “limiting”? Limiting what?

Page 2, on line 73, authors stated  “...India committed for achieving the SDG-3…” What about putting it this way:  ‘India committed to achieving the SDG-3 ‘

Page 3, on line 130, first, spell out this abbreviation: “NFHS”, what does it stand?

On page 3, the sentence on the second paragraph begins like this: “The data have been collected from different sources…”  What about putting this way: “The data was collected from different sources….”

Page 3, lines 35-36, “ we have described key variables….” Consider putting it this way: ‘we described key variable...” Or consider revising the whole sentence.

The discussion is also good, but it will be helpful to condense the discussion section. Consider making it a bit more concise and shorten the length, if possible.

Author Response

  • Comment 1: on line 20 in the abstract, it will be helpful to substitute the word “rise” with the word “increase”.

Response: As suggested by the reviewer, we have accordingly modified the sentence in the revised manuscript.

  • Comment 2: on page 2, line 60, authors stated: “However, limiting is just one of the several goals...” What do they meant by “limiting”? Limiting what?

Response: Thank you for pointing this out. By "limiting", we intended to explain the use of family planning to put off childbearing (i.e. Fertility). We have explained this sentence and, the revised text read as follows:

However, fertility limiting is just one of the several goals (e.g. timing and spacing of births and protection against sexual and reproductive tract infections, etc.) that FP accomplishes.

  • Comment 3: Page 2, on line 73, authors stated “...India committed for achieving the SDG-3…” What about putting it this way: ‘India committed to achieving the SDG-3'

Response: As suggested by the reviewer, we have accordingly modified the sentence in the revised manuscript.

  • Comment 4:Page 3, on line 130, first, spell out this abbreviation: “NFHS”, what does it stand?

Response: Thank you for pointing this out. However, the abbreviation: “NFHS” has already been mentioned on Page 3, in line 108.

  • Comment 5: On page 3, the sentence on the second paragraph begins like this: “The data have been collected from different sources…” What about putting this way: “The data was collected from different sources….”

Response: As suggested by the reviewer, we have accordingly modified the sentence in the revised manuscript.

  • Comment 6:Page 3, lines 35-36, “we have described key variables….” Consider putting it this way: ‘we described key variable...” Or consider revising the whole sentence.

Response: As suggested by the reviewer, accordingly we havemodified the sentence in the revised manuscript.

  • Comment 7: The discussion is also good, but it will be helpful to condense the discussion section. Consider making it a bit more concise and shorten the length, if possible.

Response: In most, public health papers discussion assumes an important position. We believe the discussion part is the key for nuanced arguments of the paper in comparison to the rest of the paper. Thus, we would like to keep as it is if reviewers and editor permit us.

Round 2

Reviewer 1 Report

The revised draft has addressed the reviewer's comments. 

Reviewer 2 Report

The paper looks good. All earlier suggestions addressed. Thank you!